# Sex and strategy effects on brain activation during a 3D-navigation task

Isabel Noachtar [1,2✉], Ti-Anni Harris[1,2], Esmeralda Hidalgo-Lopez [1] & Belinda Pletzer [1✉]

Sex differences in navigation have often been attributed to the use of different navigation strategies in men and women. However, no study so far has investigated sex differences in the brain networks supporting different navigation strategies. To address this issue, we employed a 3D-navigation task during functional MRI in 36 men and 36 women, all scanned thrice, and modeled navigation strategies by instructions requiring an allocentric vs. ego-centric reference frame on the one hand, as well as landmark-based vs. Euclidian strategies on the other hand. We found distinct brain networks supporting different perspectives/ strategies. Men showed stronger activation of frontal areas, whereas women showed stronger activation of posterior brain regions. The left inferior frontal gyrus was more strongly recruited during landmark-based navigation in men. The hippocampus showed stronger connectivity with left-lateralized frontal areas in women and stronger connectivity with superior parietal areas in men. We discuss these findings in the light of a stronger recruitment of verbal networks supporting a more verbal strategy in women compared to a stronger recruitment of spatial networks supporting a more spatial strategy use in men. In summary, this study provides evidence that different navigation strategies activate different brain areas in men and women.

[1] Department of Psychology and Centre for Cognitive Neuroscience, University of Salzburg, Hellbrunnerstr. 34, 5020 Salzburg, Austria. [2] These authors contributed equally: Isabel Noachtar, Ti-Anni Harris. ✉email: isabel.noachtar@sbg.ac.at; belinda.pletzer@sbg.ac.at

Although sex differences in cognition have been a matter of debate, there is a general agreement that most differences are moderate and emerge in certain sets of tasks, but not global cognition. Overall, results suggest a female advantage for verbal and memory tasks as well as face and emotion recognition (for a review see 1). Men, on the other hand, show better performance in visual-spatial tasks such as mental rotation and navigation[1–3]. Besides mental rotation, the most robust results are observed for spatial navigation for which a male advantage has been described consistently across different species[4–9] and cultures[1,3,10–14]. Various studies demonstrate that the performance differences between men and women in the navigation task depend on (i) experience or training in navigation tasks[15–17], (ii) the dimensionality of the navigation task (e.g., 2D vs. 3D)[1,13,18–22], (iii) time limits for solving the task[23], (iv) the instructions for the task/navigation strategy[2,13], and (v) hormonal factors[21,24–32]. Particularly, the female menstrual cycle[27,33], as well as the male sex hormone testosterone[31,34], have been discussed to modulate spatial performance, even though results are not unequivocal.

Despite this extensive body of behavioral research on sex differences in navigation, the neurobiological underpinnings of these sex differences are not well understood. Particularly, only few neuroimaging studies have focused on sex differences in the brain networks supporting navigation[12,14,35–37]. Grön et al. (2000) found stronger activation in left hemispheric hippocampus in men, while women showed stronger activation in the right parietal and right prefrontal cortex during a virtual maze task. Ohnishi et al. (2006) found no sex differences in brain activation during a passive-watching task, but report differences between good and poor navigators. The bilateral hippocampi were more activated for good navigators, while right parietal lobe activation correlated with poor navigational skills. In another study, sex differences for neural activation during a spatial task prevailed after controlling for performance[37]. This finding supports the assumption that men and women use neural resources differently and that this is not dependent on performance. Accordingly, the different brain activation patterns in men and women observed during navigation[35,38] may also reflect the use of different approaches to navigation.

Indeed a large body of behavioral research in both animals and humans supports the idea of differential approaches to navigation between men and women[13,38–40]. Multiple studies demonstrate that women are more likely to rely on egocentric navigation than men, while men are more likely to rely on allocentric navigation than women[10,13,41–44]. Egocentric navigation is determining the directions relative to the own position (left, right, straight ahead), whereas allocentric navigation determines directions using a framework independent of the own position (north, south, east, west)[45,46]. In the current study, we use the term "perspective" to distinguish between allocentric and egocentric reference frames (compare also[2,47]). Furthermore, women show a stronger preference for landmark information than men[13,20,42–44,48,49], particularly when forced to rely on an allocentric reference frame[2]. Men on the other hand prefer distance descriptions in Euclidian terms (meters/miles)[13,20,42–44,48,49]. In the current study, we use the term "strategy" to distinguish between Euclidian and landmark-based navigation. It has been speculated that these differential approaches to navigation may result from differences in the perception of the environment[50,51] and may underlie—at least in part—the sex differences in navigation ability[39,52]. However, navigation strategies have not been accounted for in previous neuroimaging studies on sex differences in navigation. Therefore, it is unclear, whether differential activation patterns observed in men and women during navigation are attributable to differential navigation strategies—utilizing different brain networks—or simply indicate a differential allocation of effort/neural resources to different aspects of a task.

Accordingly, the present study was designed to better understand how different navigation strategies contribute to sex differences in brain activation during navigation on the one hand and the neural substrates underlying these sex differences in navigation strategies on the other hand. Thus, we aimed to compare the brain activation of men and women during navigation when they were not free to choose their preferred approach to navigation, but their navigation strategy was predetermined by the directions given. To achieve this situation, we employed a 3D-navigation task, previously developed to (i) modulate perspective and strategy via different instructions in a $2 \times 2$ design, and (ii) closely model modern real-world navigation in a large sample of 36 men and 36 women during fMRI (functional magnetic resonance imaging). By scanning each subject three times, we were also able to control for potential learning and menstrual-cycle effects, which have been published previously[53]. Importantly, while overall activation patterns are cycle-dependent, no strategy-dependent shifts in brain activation were observed along the menstrual cycle[53].

Since animal studies clearly pinpoint different navigation strategies to certain brain areas, we combine a region-of-interest (ROI) based approach with exploratory whole-brain analyses. In the animal literature, the so called spatial (i.e., allocentric, landmark-based) strategy is considered hippocampus-dependent[54] (53, 57), while the so called stimulus-response (i.e., egocentric) strategy shows stronger involvement of the caudate[55–57]. Also in humans the hippocampus has been associated with a landmark-based strategy[58,59], and the encoding of the spatial relationship between stimuli as required during allocentric navigation[45]. Another brain area involved in landmark-based navigation is the retrosplenial cortex[60], which is also relevant for the integration of reference frames during allocentric navigation[61,62]. Consistently, lesion studies found retrosplenial pathology linked to spatial disorientation[63]. Therefore, we will focus on hippocampus, caudate and retrosplenial cortex for our ROI-based analyses. Given that the hippocampus and retrosplenial cortex are relevant to both allocentric navigation, which is preferred by men, and landmark-based navigation, which is preferred by women, we refrain from directional hypotheses regarding sex differences in the activation of these areas at this point. However, we do seek to further explore the differential involvement of these brain areas in different aspects of navigation by using them as seeds for functional connectivity analyses. In order to gain an idea, whether sex differences in connectivity patterns are navigation specific or may already reflect perceptional differences, we added the primary visual cortex (V1) as a seed to our connectivity analyses. Furthermore, we explore sex differences in the effects of perspective and strategy on overall brain activation patterns using whole-brain analyses.

## Results

**Behavioral results**. The behavioral results showed that men reached significantly more targets than women irrespective of instruction ($b = 0.77$, $SE_b = 0.16$, $t_{(69)} = 4.84$, $p < .001$). Furthermore, participants reached significantly more targets with egocentric compared to allocentric instructions ($b = 0.71$, $SE_b = 0.03$, $t_{(781)} = 24.74$, $p < .001$) and with landmark-based compared to Euclidian instructions ($b = 0.09$, $SE_b = 0.03$, $t_{(781)} = 2.96$, $p < 0.01$; compare Table 1) irrespective of sex.

**ROI-based analyses**. None of the sex differences in activation from the selected ROIs survived multiple comparison correction of the p-value (all $|b| < 0.44$, all $|t| < 2.00$, all $p > 0.05$). There was

**Table 1 Performance and BOLD-response in the hippocampus, caudate, and retrosplenial cortex separated by sex and perspective.**

**Allocentric**

**Euclidian**

| Sex | Nr. hits | BOLD-response | | | | | | |
|---|---|---|---|---|---|---|---|---|
| | | Hippocampus | | Caudate | | Retrosplenial Cortex | | |
| | | Left | Right | Left | Right | Left | Right | |
| Women | | | | | | | | |
| Mean | 2.74 | −0.22 | −0.25 | −0.11 | −0.12 | −0.41 | −0.44 | |
| SD | 0.74 | 0.25 | 0.32 | 0.33 | 0.36 | 0.39 | 0.40 | |
| Men | | | | | | | | |
| Mean | 3.53 | −0.18 | −0.18 | −0.03 | −0.07 | −0.54 | −0.66 | |
| SD | 0.86 | 0.25 | 0.30 | 0.37 | 0.39 | 0.39 | 0.39 | |

**Landmark**

| Sex | Nr. hits | BOLD-response | | | | | | |
|---|---|---|---|---|---|---|---|---|
| | | Hippocampus | | Caudate | | Retrosplenial Cortex | | |
| | | Left | Right | Left | Right | Left | Right | |
| Women | | | | | | | | |
| Mean | 2.84 | −0.18 | −0.23 | −0.11 | −0.13 | −0.39 | −0.43 | |
| SD | 0.76 | 0.27 | 0.33 | 0.36 | 0.39 | 0.39 | 0.39 | |
| Men | | | | | | | | |
| Mean | 3.52 | −0.18 | −0.20 | −0.08 | −0.08 | −0.50 | −0.63 | |
| SD | 0.95 | 0.30 | 0.35 | 0.42 | 0.44 | 0.37 | 0.38 | |

**Egocentric**

**Euclidian**

| Sex | Nr. hits | BOLD-response | | | | | | |
|---|---|---|---|---|---|---|---|---|
| | | Hippocampus | | Caudate | | Retrosplenial Cortex | | |
| | | Left | Right | Left | Right | Left | Right | |
| Women | | | | | | | | |
| Mean | 3.43 | −0.21 | −0.21 | −0.12 | −0.15 | −0.48 | −0.50 | |
| SD | 0.75 | 0.27 | 0.34 | 0.40 | 0.43 | 0.40 | 0.39 | |
| Men | | | | | | | | |
| Mean | 4.14 | −0.17 | −0.15 | −0.06 | −0.11 | −0.59 | −0.69 | |
| SD | 0.90 | 0.30 | 0.34 | 0.40 | 0.39 | 0.41 | 0.41 | |

**Landmark**

| Sex | Nr. hits | BOLD-response | | | | | | |
|---|---|---|---|---|---|---|---|---|
| | | Hippocampus | | Caudate | | Retrosplenial Cortex | | |
| | | Left | Right | Left | Right | Left | Right | |
| Women | | | | | | | | |
| Mean | 3.54 | −0.19 | −0.22 | −0.15 | −0.19 | −0.42 | −0.43 | |
| SD | 0.63 | 0.27 | 0.34 | 0.37 | 0.39 | 0.40 | 0.40 | |
| Men | | | | | | | | |
| Mean | 4.26 | −0.14 | −0.14 | −0.09 | −0.14 | −0.53 | −0.63 | |
| SD | 0.91 | 0.30 | 0.34 | 0.42 | 0.42 | 0.41 | 0.43 | |

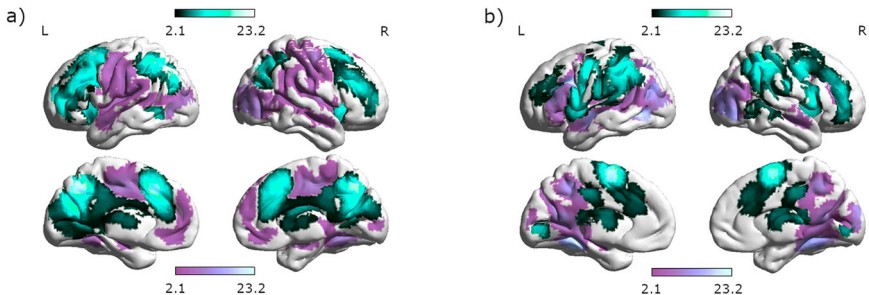

**Fig. 1 Modulation of brain activation by perspective and strategy. a** Differential activation between allocentric and egocentric navigation. Areas with stronger activation for allocentric compared to egocentric navigation are displayed in turquoise. Areas with stronger activation for egocentric compared to allocentric navigation are displayed in purple. **b** Differential activation between Euclidian-based and landmark-based navigation. Areas with stronger activation for Euclidian compared to landmark-based navigation are displayed in turquoise. Areas with stronger activation for landmark-based compared to Euclidian navigation are displayed in purple.

also no significant difference in hippocampus, caudate, or retrosplenial cortex activation between the allocentric and egocentric perspective (all $|b| < 0.11$, all $|t| < 2.13$, all $p > 0.05$) or the landmark-based and Euclidian strategy (all $|b| < 0.11$, all $|t| < 2.49$, all $p > 0.05$). All ROIs were deactivated irrespective of instruction or sex (compare Table 1).

**Whole-brain analyses**. Overall, the navigation task activated a large fronto-parietal network with deactivations in default mode areas.

**Effect of perspective**. Allocentric navigation was accompanied by stronger activation in a large bilateral network consisting of the bilateral superior frontal gyri, supplementary motor cortex (SMC), bilateral cingulate gyri as well as left superior parietal lobe, left angular gyrus, and bilateral precuneus compared to egocentric navigation (Fig. 1a). Stronger activation during egocentric compared to allocentric navigation was observed bilaterally in medial and superior frontal cortex, left precentral, postcentral and middle cingulate gyri, right superior parietal lobe, bilateral inferior occipital gyri, bilateral lingual and fusiform gyri, and left anterior insula (Supplementary Table 1).

**Effect of strategy**. Euclidian navigation was accompanied by stronger activation in the bilateral precentral gyri, left middle frontal gyrus, bilateral middle and posterior cingulate gyri, bilateral SMC, and right supramarginal gyrus than landmark-based navigation (Fig. 1b). Landmark-based navigation was accompanied by stronger activation in the bilateral inferior frontal gyri (IFG) and bilateral orbital gyri, bilateral temporal gyri as well as bilateral fusiform and parahippocampal gyri and right anterior and left posterior insula compared to Euclidian navigation (Supplementary Table 2).

**Interactive effects between perspective and strategy**. Interactions between perspective and strategy were observed in the bilateral lingual gyri (left: $[-9, -76, -2]$, 44 voxels, T = 5.53, $p_{FWE} = 0.001$; right: $[18, -91, 1]$, 47 voxels, T = 8.81, $p_{FWE} < 0.001$). In contrast to the egocentric perspective, the allocentric condition showed marked differences between the Euclidian and landmark condition (Fig. 2b): The left hemisphere was less deactivated for the Euclidian compared to the landmark condition in the allocentric perspective. The opposite pattern was observed in the right hemisphere for the allocentric perspective only (Fig. 2a).

**Sex differences**. Larger activation in women compared to men was seen bilaterally in the pre- and postcentral gyri, right

posterior middle frontal gyrus and bilateral middle occipital gyri, right supramarginal and superior parietal gyri, as well as right cuneus (Fig. 3). In comparison, larger activation in men compared to women were found in left superior and medial frontal gyri, bilateral supplementary motor cortex, left pre- and postcentral areas, bilateral anterior and middle cingulate gyri, as well as right frontal and left rolandic operculum and left putamen (Supplementary Table 3). Sex differences in brain activation were not mediated by sex hormone effects.

**Interaction between sex and perspective/strategy**. No interaction between sex*perspective was observed. A significant interaction of sex*strategy was observed in the left inferior frontal gyrus (opercular part) ($[-39, 14, 22]$, 159 voxels, T = 4.83, $p_{FWE} = 0.019$) (Fig. 4a). In this area men showed significantly stronger activation for landmark-trials compared to women, while no differences in activation for Euclidian trials were observed (Fig. 4b).

**Connectivity analysis: effects of perspective**. Both, the left and the right hippocampus showed stronger connectivity for allocentric compared to egocentric navigation with the middle cingulate cortex, as well as with the right supramarginal gyrus (Fig. 5a, b). The right hippocampus additionally showed stronger connectivity for allocentric compared to egocentric navigation with the bilateral insula and left fusiform gyrus.

Both, the left and right hippocampus showed stronger connectivity for egocentric compared to allocentric navigation with the bilateral precuneus. Additionally, the right hippocampus showed stronger connectivity for egocentric compared to allocentric navigation with the right middle frontal gyrus, bilateral angular gyri, and left middle occipital gyrus (Supplementary Tables 4, 5).

Both, left and right caudate, as well as the left retrosplenial cortex showed stronger connectivity for allocentric compared to egocentric navigation with the left middle frontal and bilateral superior parietal cortex as well as with the bilateral temporal and occipital lobe and putamen (Fig. 5c–e). The left caudate showed stronger connectivity during allocentric navigation compared to egocentric perspective with the left angular gyrus, whereas the right caudate showed stronger connectivity with the right angular gyrus. The right retrosplenial cortex showed stronger connectivity with the right precuneus for allocentric compared to egocentric perspective (Fig. 5f).

Left and right caudate, as well as the left retrosplenial cortex showed stronger connectivity for egocentric compared to allocentric navigation with the right frontal and bilateral posterior orbital gyri, supplementary motor cortex, right pre- and

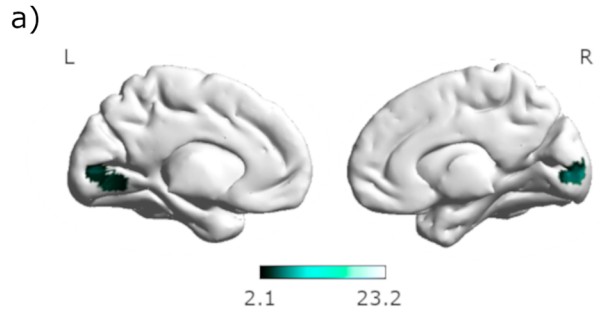

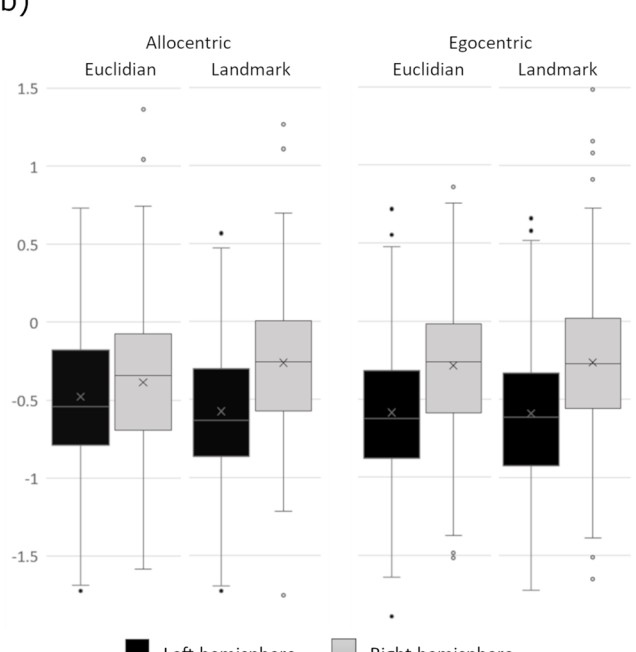

**Fig. 2 Interactive effect between perspective and strategy in the lingual gyrus. a** Differential activation of the left and right hemisphere in the lingual gyrus. **b** In the left hemisphere, Euclidian navigation yielded less deactivation than landmark navigation for allocentric compared to egocentric instructions. In the right hemisphere, the opposite pattern was observed. Error bars represent standard errors, $N = 72$.

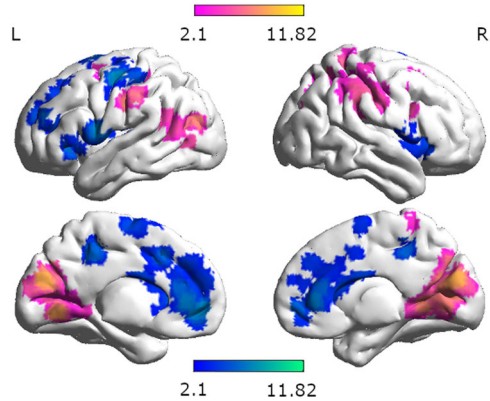

**Fig. 3 Differential brain activation during navigation in women and men.** Pink areas indicate stronger activation in women compared to men. Blue areas indicate stronger activation in men compared to women.

$p_{\text{FWE}} = 0.005$; right: [15, 11, 13], 112 voxels, T = 4.95, $p_{\text{FWE}} = 0.030$). The left and right retrosplenial cortex showed stronger connectivity for Euclidian compared to landmark-based navigation to the right lateral occipital cortex and stronger connectivity for landmark-based compared to Euclidian navigation with left precentral gyrus, left supplementary motor cortex, putamen, and left lingual gyrus (compare Supplementary Tables 12, 13).

For Euclidian compared to landmark the primary visual cortex showed stronger connectivity to occipital areas. During landmark navigation the primary visual cortex showed stronger connectivity with ipsilateral frontal and other cortical areas (Supplementary Tables 14, 15). No interactions between perspective and strategy were observed in connectivity of any ROI.

**Sex differences.** Both left and right hippocampus, showed stronger connectivity for women compared to men with the left frontal cortex, angular and temporal gyri, and bilateral posterior cingulate gyri (Fig. 6a, b). A similar pattern was observed for connectivity of the left and right retrosplenial cortex, with weaker connectivity to frontal areas, but stronger connectivity to the precuneus compared to the hippocampus (Fig. 6e, f). In addition, the left and right hippocampi showed stronger connectivity with each other in women compared to men. For men compared to women left and right hippocampus showed stronger connectivity with the left medial frontal cortex and bilateral superior parietal lobes. Furthermore, the left hippocampus showed stronger connectivity for men compared to women with right frontal and precentral areas (Supplementary Tables 16, 17). Again, a similar pattern was observed for the connectivity of the left and right retrosplenial cortex. In addition men showed stronger connectivity between the retrosplenial cortices and occipital areas than women (Supplementary Tables 18, 19).

Both the left and right caudate showed stronger connectivity for women than men with each other, as well as with the left temporal gyrus, planum polare, and bilateral thalamus (Fig. 6c, d). For men compared to women left caudate showed stronger connectivity with the right superior parietal lobule and the right caudate was stronger connected with the right precentral gyrus (Supplementary Tables 20, 21). Sex differences in brain connectivity were not mediated by sex hormone effects.

The primary visual cortex showed stronger connectivity with early visual areas in women compared to men (Fig. 6g, h). For men compared to women, the primary visual cortex showed stronger connectivity with higher visual areas in the occipital gyrus, with the left angular gyrus, as well as right frontal and temporal areas (Supplementary Tables 22, 23).

postcentral gyri, bilateral cingulate and fusiform gyri and the parietal operculum (Supplementary Tables 6, 7).

Interestingly, the left retrosplenial cortex showed stronger connectivity to the anterior insula for allocentric compared to egocentric navigation, while the left caudate showed stronger connectivity to the anterior insula for egocentric compared to allocentric navigation (Supplementary Tables 8, 9).

A modulation by perspective was already observed in connectivity of primary visual areas (Fig. 5g, h). The bilateral primary visual cortex showed stronger connectivity with mostly left-lateralized middle frontal, middle temporal and inferior parietal areas during allocentric compared to egocentric navigation (Supplementary Tables 10, 11).

**Effects of strategy**. No strategy effects were observed for connectivity of the left or right hippocampus and the left caudate. The right caudate showed stronger connectivity for Euclidian compared to landmark-based navigation with the left superior occipital gyrus ([−12, −88, 34], 142 voxels, T = 5.09, $p_{\text{FWE}} = 0.015$) and bilateral putamen (left: [−15, 11, −2], 88 voxels, T = 5.30,

a)

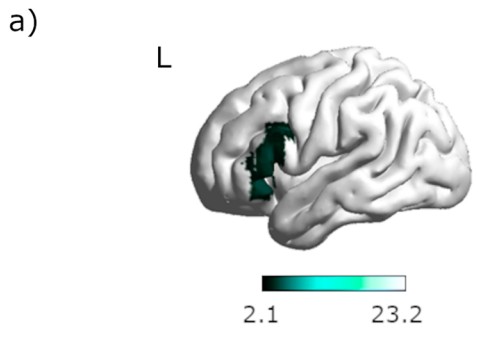

L

b)

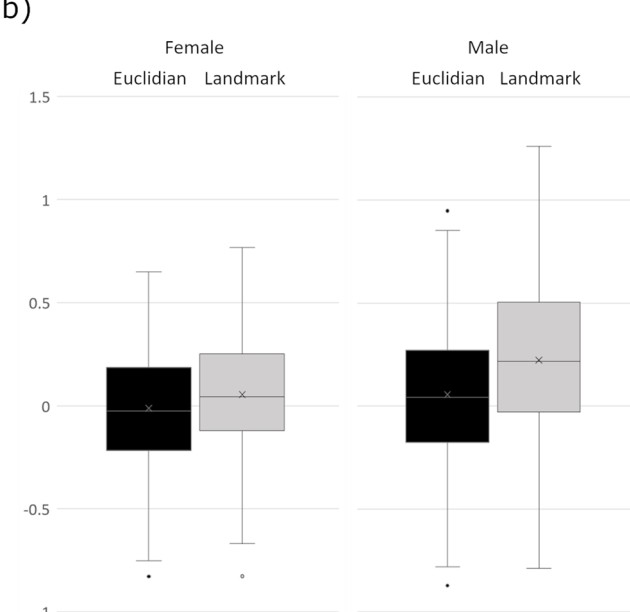

**Fig. 4 Interaction between sex and strategy in the left inferior frontal gyrus. a** Differential activation for men during landmark compared to Euclidian navigation in the left inferior frontal gyrus. **b** Men showed significantly stronger activation during landmark navigation, but not during Euclidian navigation. Error bars represent standard errors, $N = 72$.

**Interaction between sex and perspective/strategy**. We observed a significant interaction between sex and perspective for the connectivity between the right caudate and the right superior frontal gyrus ([15, 11, 70], 53 voxels, T = 4.88, $p_{FWE} = 0.044$) (Fig. 7a). Women showed stronger negative connectivity during allocentric compared to egocentric navigation, whereas men showed greater negative connectivity during egocentric compared to allocentric navigation (Fig. 7b).

No interaction between sex and strategy was observed for the connectivity of any of the ROIs and sex hormones did not affect brain activation or connectivity during the navigation task.

## Discussion

The goal of the current manuscript was to investigate sex differences in brain activation and connectivity during navigation when the perspective and strategy utilized by men and women were predetermined by the directions given. This approach allows us to differentiate brain activations and connectivity patterns that differ between men and women during navigation irrespective of perspective/strategy from brain activation and connectivity patterns that only differ when certain navigation strategies are required. Noteworthy, all brain areas were either activated or deactivated during the task and even the connectivity of primary visual areas was modulated by sex and task conditions. Accordingly, the overall sex differences described in the current manuscript may not be specific to navigation but may represent a more general pattern of functional connectivity differences guiding perception, that predispose subjects to process the spatial materials presented in the navigation task in different ways.

As a first observations guiding the following discussion it should be noted, that all regions of interest selected for their suspected involvement in navigation, were deactivated during the task across the different conditions. Although we cannot completely rule out that participants were engaging these areas during the baseline condition (blank screen), we think that the following alternative explanation is more likely. While these areas are commonly activated in traditional navigation tasks, where participants have to remember their surroundings[55–60], the hippocampus in particular is commonly deactivated during various working-memory tasks[64,65], particularly during high working-memory load[66]. Our task was designed to reflect modern real-world navigation, which commonly involves finding a way through a novel environment with the help of some sort of navigation system providing directions. For this type of task it is less relevant to encode or retrieve spatial information from long-term memory, but rather to continuously update and manipulate new spatial information encountered in the environment. Accordingly, this task—like modern real-world navigation—has a stronger emphasis on spatial working memory rather than spatial learning, which is reflected in the observed (de-)activation patterns. Thus, effects of perspective and strategy in this task point towards how different types of information are updated/manipulated during spatial working memory, rather than encoded during spatial learning. Nevertheless, the connectivity patterns of the hippocampus may well indicate from which areas the hippocampus preferably retrieves the information to encode when encountering a new environment, even if encoding may not be strictly necessary.

This study corroborates former studies in showing (i) a male advantage in navigation performance (irrespective of perspective and strategy), and (ii) distinct brain networks for different types of navigation. In addition, we found (iii) stronger frontal activation in men and stronger posterior activation in women, (iv) an interactive effect of sex and strategy on activation in the left IFG, (v) stronger connectivity of the hippocampus and left retrosplenial cortex with left-lateralized frontal areas in women, and stronger connectivity of the hippocampus and left retrosplenial cortex with superior parietal areas in men. In the following, these results will be discussed in further detail.

Regarding different types of navigation, two results are of interest. First, it has to be noted that behavioral results of the present study demonstrate the expected main effects of perspective, strategy, and sex, but—unlike previous studies[2,13,47]—do not demonstrate any interaction between sex and perspective or strategy. However, the task utilized in the present study, was optimized for the assessment of brain activation and thus substantially modified from previous behavioral versions. More specifically, in previous behavioral versions a fixed target had to be reached with no time limit and navigation time was assessed as a measure of performance. In the current fMRI-adaptation, participants could reach a series of targets within a given time limit and the measure of performance was the number of targets reached. This target number is a discrete variable and shows less variability than navigation time. Thus, our current performance measure may not be as sensitive to interactive effects as the measure used in previous behavioral studies. The missing performance interaction between sex and perspective/strategy could be also due to the MRI situation: Participants lie in a noisy,

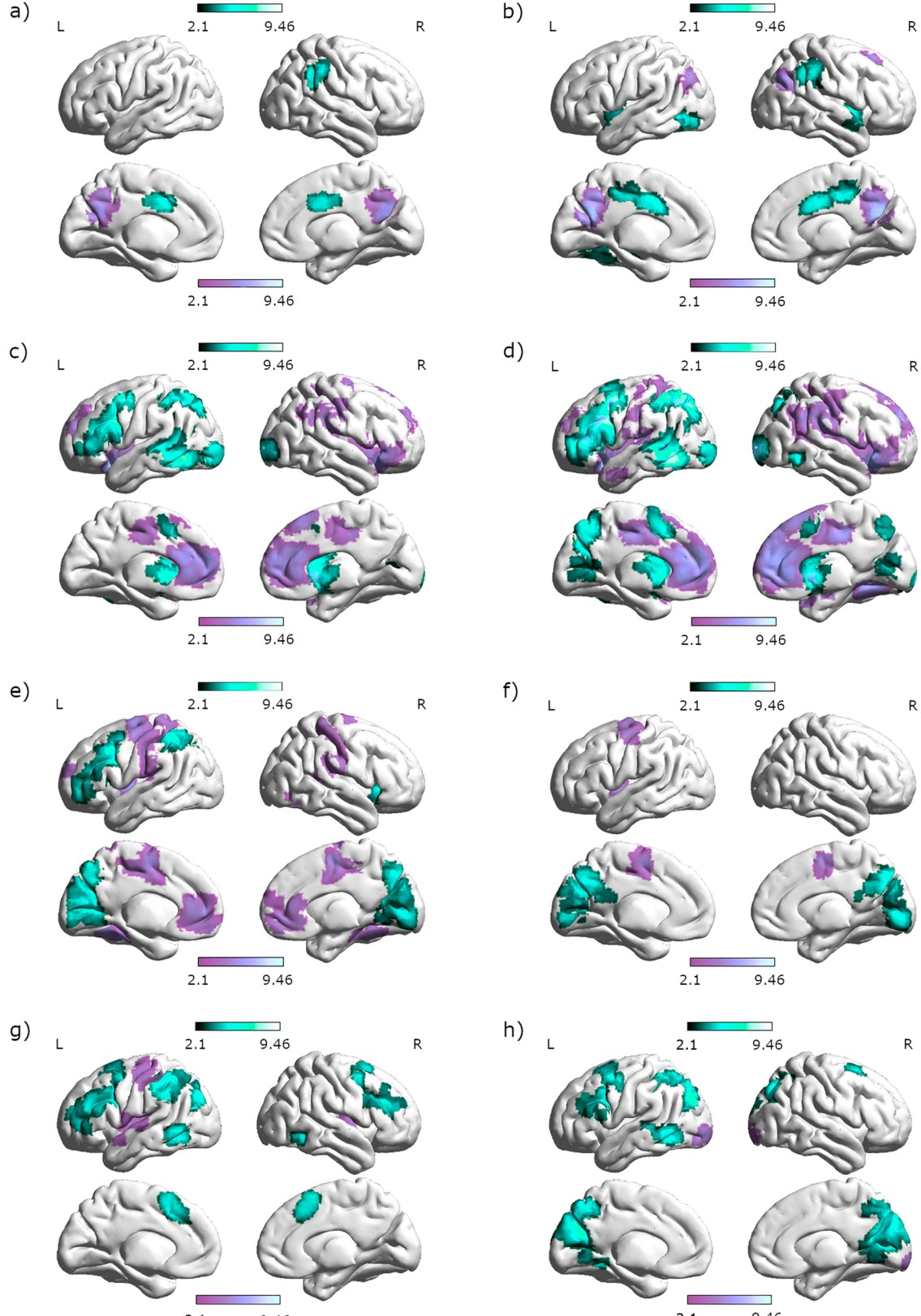

**Fig. 5 Areas with differential connectivity during allocentric and egocentric navigation.** Areas with stronger connectivity for allocentric compared to egocentric navigation are displayed in turquoise. Areas with stronger connectivity for egocentric compared to allocentric navigation are displayed in purple. Seed regions: **a** left hippocampus, **b** right hippocampus, **c** left caudate, **d** right caudate, **e** left retrosplenial cortex **f** right retrosplenial cortex, **g** left primary visual cortex, **h** right primary visual cortex.

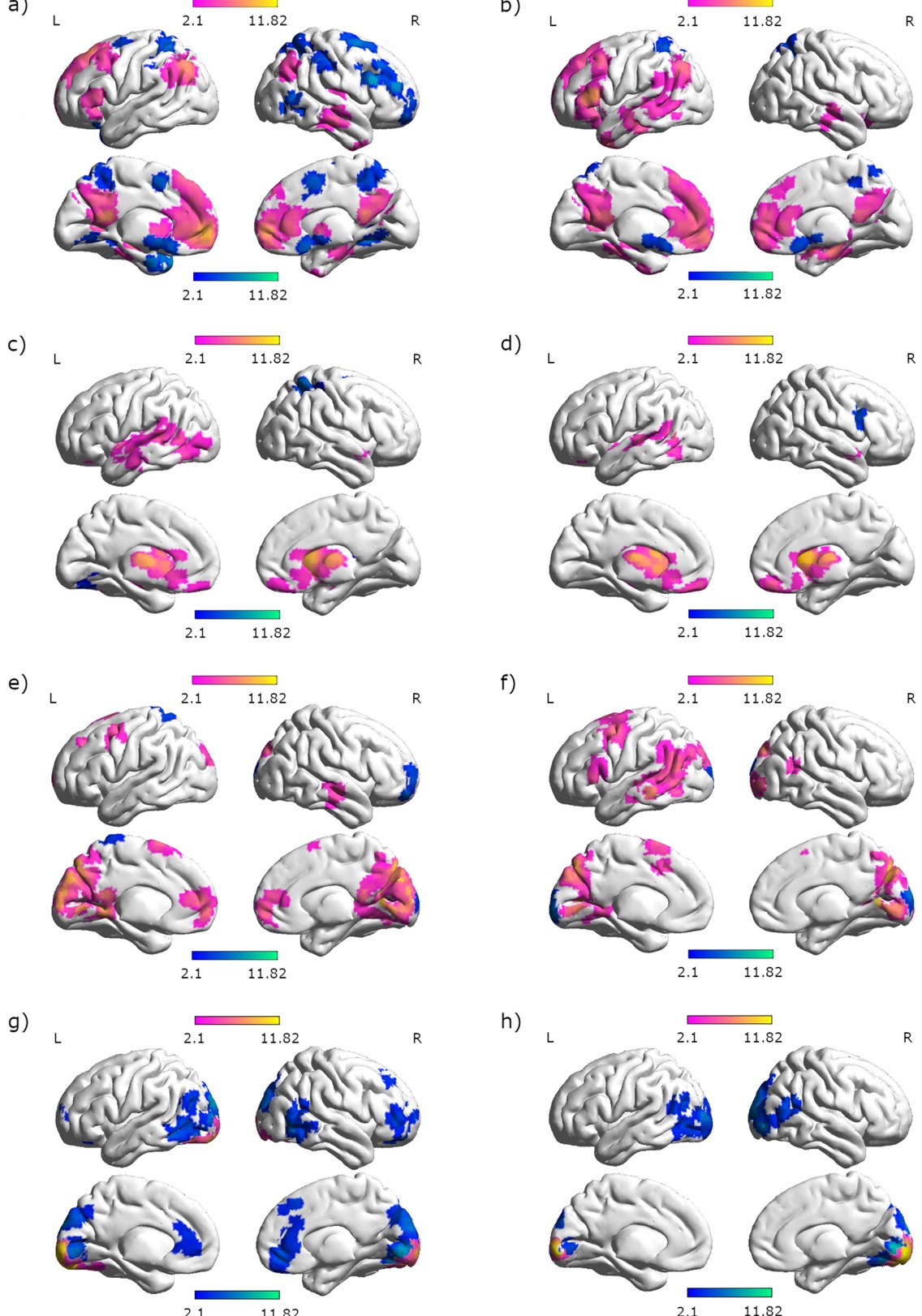

**Fig. 6 Areas with differential connectivity between men and women.** Areas with stronger connectivity for women compared to men are displayed in pink. Areas with stronger connectivity for men compared to women are displayed in blue. Seed regions: **a** left hippocampus, **b** right hippocampus, **c** left caudate, **d** right caudate, **e** left retrosplenial cortex, **f** right retrosplenial cortex, **g** left primary visual cortex, **h** right primary visual cortex.

a)

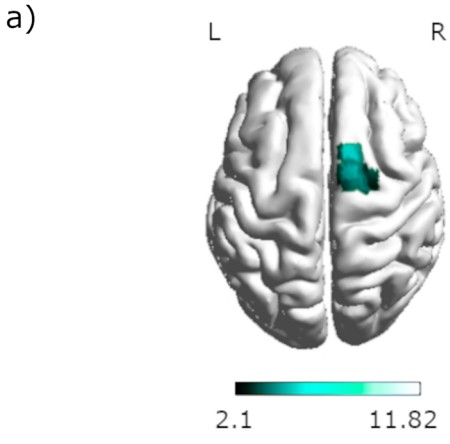

b)

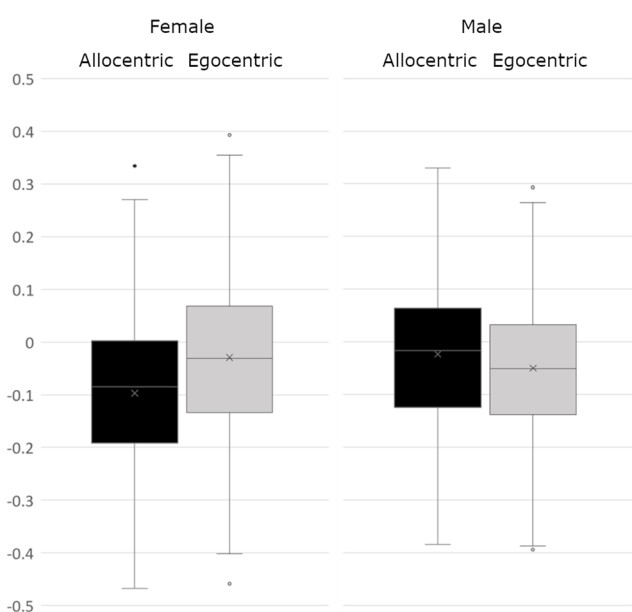

**Fig. 7 Significant interaction between sex and perspective in the right superior frontal gyrus. a** Differential connectivity between the right caudate and right superior frontal gyrus in women compared to men. **b** Females showed stronger negative connectivity for allocentric compared to egocentric navigation, whereas men showed greater negative connectivity during egocentric than allocentric navigation. Error bars represent standard errors, N = 72.

narrow scanner and experience different spatial reference frames than during usual upright navigation[67]. Outside the scanner, time constraints lead to sex differences in navigation performance[23]. Inside the MRI scanner, however, the time constraints may mask the reported sex differences for perspective/strategy performance due to the stressful situation with subsequent increased reaction time[68].

Second, our neuroimaging results clearly indicate that perspective and strategy, even though often confounded in behavioral tasks[13], show distinct brain activation patterns (compare Fig. 1). The following observations stand out regarding the differential activation patterns during different perspectives/strategies: (i) Allocentric navigation shows stronger involvement of frontal and parietal areas than egocentric navigation. This is likely attributable to the need to transform one reference frame into another and imaginary rotate oneselves in order to decide which direction to go in the allocentric condition. This interpretation is

supported by studies showing activation in the frontal lobe and left parietal cortex during self-rotation in relation to fixed objects[62] and body-centered judgements relative to external cues[69].

(ii) Allocentric navigation is associated with stronger parietal activation, while egocentric navigation is associated with stronger temporal activation. This observation is in line with the dual path model, suggesting that movements and object location are processed along the dorsal visual stream, extending from the occipital cortex to the inferior parietal cortex, while object identity is processed along the ventral visual stream, extending from the occipital cortex to the inferior temporal cortex[70]. (iii) The fronto-parietal network shows lateralized connectivity with the caudate, which recruits left frontal and parietal areas more strongly during allocentric navigation, whereas right frontal and parietal areas are more activated during egocentric navigation. This is in line with a former study in healthy participants showing mainly right posterior parietal and frontal premotor activation during a computation task with an egocentric reference frame[71].

(iv) Landmark navigation shows stronger involvement of inferior frontal and left temporal areas. This observation may reflect the need for verbal labelling and memorization of the landmarks[72]. Indeed our IFG coordinates [−39, 14, 22] were most commonly associated with semantic processing (Z = 9.68, posterior probability 0.75), language processing (Z = 8.11, posterior probability 0.72), phonological processing (Z = 7.24; posterior probability 0.76), and word processing (Z = 6.84, posterior probability = 0.70) on the neurosynth database (www.neurosynth.org). Vice versa, the fact that Euclidian navigation shows stronger involvement of superior parietal and superior frontal areas may reflect the need for numerical processing[73]. Interestingly, navigation strategy is not reflected in connectivity patterns of the subcortical ROIs.

Finally, an interaction between perspective and strategy was observed in the bilateral lingual gyrus, which is considered an important navigation area in humans[35,74], particularly involved in the encoding and retrieval of landmark information (buildings/landscapes)[75,76]. While the discrimination of different landmarks is of particular relevance to the landmark condition of the current task, the nature of the task does not require the encoding or retrieval of landmark information. During allocentric navigation, the right lingual gyrus was less deactivated in the landmark condition compared to the Euclidian condition, while the opposite pattern was observed in the left lingual gyrus. This suggests a stronger lateralization of lingual gyrus activation during navigation conditions in which landmark information is particularly relevant. On the one hand, lateralization has often been discussed as a mechanism to increase processing efficiency[77]. On the other hand, the additional recruitment of homotopic contra-lateral brain areas has often been described as an efficient mechanism to handle increased working-memory load[78,79]. The current results are in accordance with (i) lateralized recruitment of the lingual gyrus in conditions where landmark information is relevant, and (ii) more bilateral processing in the condition with the highest working-memory load, i.e., the allocentric-Euclidian condition.

Irrespective of perspective or strategy, women showed stronger activation in posterior areas, including occipital, temporal and parietal lobes, whereas men showed stronger activation in left-lateralized frontal areas. These results are in line with another study, demonstrating greater bilateral frontal and right precentral activation in men compared to women in a spatial memory task[14]. Furthermore, stronger frontal and cingulate activation during the encoding phase of a virtual maze task has been shown in men compared to women[37]. Our result of stronger central and parietal activation in women is supported by a former study showing stronger involvement of paracentral regions and the right parietal gyrus in women compared to men[37].

Accordingly, while the differential brain networks utilized by men and women in other spatial tasks have been interpreted as indicating differential strategy use, this may not necessarily be the case in the present navigation task, since the results occurred irrespective of the predetermined perspective or strategy. Instead, the observed sex differences may be more easily explained by the allocation of neuronal resources to those aspects of the task that are perceived as more effortful. For instance, the increased left frontal activation in men appears to be particularly strong during landmark navigation as indicated by a significant interaction between sex and strategy in the left IFG. If the IFG activation is indeed indicative of verbal processing as suggested by the neurosynth analysis, these results suggest that the verbal processing required during landmark navigation is more effortful in men. This idea is in line with the repeated observation of a female preference and advantage for landmark navigation[2,80,81]. It has repeatedly been demonstrated that the availability of landmark information reduces performance differences between men and women[2,13,18,19,81]. During 2D-navigation, women perform better than men when directions are phrased from an egocentric perspective using landmark terms[13]. Vice versa, the stronger posterior activation observed in women might be indicative of visuo-spatial processing being more effortful in women, irrespective of the task instructions. The attempt to minimize cognitive effort may in turn explain, why different approaches to navigation are chosen by men and women, when perspective and strategy are not predetermined. This notion is also supported by our finding of stronger connectivity between the hippocampus and superior parietal areas in men, suggesting a more efficient network for visuo-spatial memory formation in men compared to women. Indeed, the areas preferentially recruited by the hippocampus in men were most commonly associated with action observation and spatial attention[82]. Vice versa, the hippocampus recruits temporal and left frontal areas more strongly in women, which may suggest again a more efficient network for verbal memory formation in women. Interestingly, a very similar pattern of connectivity differences between men and women was observed for the retrosplenial cortex, an area involved in the encoding and retrieval of landmark information. It can be speculated, that the verbal labelling of landmarks place a stronger role for memory formation in women, while the spatial position of landmarks plays a stronger role for memory formation in men. Also in line with this idea, is the observation that the caudate connects to left temporal areas more strongly in women compared to men, since connectivity between left caudate and left temporal gyrus was found to be particularly strong during the encoding phase of a verbal working-memory task[83]. Finally, the significant interaction of sex and perspective in Fig. 7 may indicate a stronger top-down inhibition of the caudate during the allocentric condition in women, but a stronger top-down inhibition of the caudate during the egocentric condition in men. A speculative interpretation may be that women have to suppress their preferred strategy/reference frame during the allocentric condition, while men have to suppress their preferred strategy/reference frame during the egocentric condition. This interpretation is supported by a previous study showing that cognitive control of response interference is facilitated by a fronto-striatal circuitry. The caudate was found to have a contribution to the selective inhibition of interfering response tendencies[84].

In summary, we were able to identify sex differences in brain activation patterns across different navigation conditions. Our results suggest that differential strategy preferences between men and women observed in previous behavioral studies may be the result of differential recruitment of neuronal resources, in particular differential hippocampal connectivity in men and women. While the results were irrespective of activational effects of sex hormones, they do not allow any conclusions on whether the observed sex differences are the result of organizational effects of sex hormones or socialization. The stronger hippocampal recruitment of language areas in women may support navigation strategies or spatial working-memory strategies based on the labeling of landmarks. The stronger hippocampal recruitment of areas involved in spatial attention may support navigation strategies based on reference frame changes, evaluating distances for Euclidian cues or cognitive mapping. It has also been suggested that cognitive strategies in women rely on stronger processing of visual information along the ventral stream, while cognitive strategies in men rely on stronger processing of visual information along the dorsal stream[85]. The current results are in line with this model.

## Method

**Participants**. A total of 72 healthy participants (36 men and 36 women) with an age range from 20 to 34 years (men: mean age = 25.83 years, SD = 3.35 years; women: mean age = 26.39 years, SD = 4.35 years) with no significant difference in age between men and women ($t_{(70)} = 0.60$, $p = .55$) were scanned thrice for this MRI-study. Exclusion criteria were physical, endocrine and mental illness, hormonal contraception or medication, and left-handedness. Participants signed an informed consent, in which all requirements were listed and explained. All participants had a minimum of 9 years of secondary education and had passed general qualification for university entrance ($n = 34$, 47.2%) or had a university degree ($n = 24$; 33.3%). We used the Advanced Progressive Matrices (APM)[86] to evaluate the IQ of each participant. There was no significant difference in average IQ ($t_{(67)} = 1.20$, $p = .23$) between men ($M = 107.88$, $SD = 11.19$) and women ($M = 110.89$, $SD = 9.61$). Furthermore, women were naturally cycling and had regular menstrual cycle, with a length ranging from 25 to 35 days ($M = 29.03$; $SD = 2.71$).

**Ethics statement**. The University of Salzburg's ethics committee approved the experiment, which was also in accordance with the Code of Ethics of the World Medical Association (Declaration of Helsinki). Written consent was given by all participants. A subject ID (VP001, VP002, etc.) was assigned to all participants upon arrival at the lab, which was used throughout the study to ensure anonymity.

**Navigation task**. The navigation task used for this study was made with the Unreal Engine 4 Version 12, and is an MRI-adaptation of the 3D-Navigation task used by Harris et al. (2019), which originated from Saucier et al. (2002). The checkerboard environment consisted of 100 squares (10 × 10) with an additional starting square positioned on the outside. One of 10 items (tree, bridge, stairs, house, church, bench, boulder, street light, fence, flowers) was placed in each square. Each item only appeared once in each row and column, meaning each item could be found ten times in each level (Fig. 8a). The participants used the four-button box of the MRI to navigate through the virtual world. One line of instruction was presented on screen, upon arrival at the target location the next line of instruction was presented and so on, until the 30 second time frame expired. The task was to reach as many target locations as possible within the 30 seconds. This 30 second time frame is different from the open ended time frame in the behavioral task[2], since navigation time varied drastically between individuals making MRI scan incomparable.

Four conditions were characterized by different phrasings of the directions, which modulated perspective (allocentric or egocentric) and strategy (landmark or Euclidian) in a 2 × 2 design: *allocentric + Euclidian* ("go east for 4 blocks"), *allocentric + landmark* ("go east until you reach the tree"), *egocentric + Euclidian* ("turn right and go for 4 blocks"), and *egocentric + landmark* ("turn right and go until you reach the tree"). Participants underwent four training levels with each type of directions appearing once. In the task itself, four levels represented each condition, resulting in a total of 16 levels. The order of conditions was pseudo-randomized. The starting and facing cardinal direction was only given once at the beginning of each level on the starting field (Fig. 8b). The starting field was positioned on each edge of the map four times, meaning each participant started facing the world from north, east, south, and west four times. Each path encompassed a maximum of 45 squares and 3 targets covered 15 squares. Starting direction and number of squares before a turn were counterbalanced across conditions. The participants advanced to the next level when the 30 second limit expired. As measures of performance, the number of target locations reached (Targets Reached—TR) was recorded. It was assumed that the number of successfully reached target locations reflected navigation abilities.

**Procedure**. Participants completed the 3D-navigation task as part of a larger MRI/MEG study. During a pretest at the Faculty of Natural Sciences of the University of Salzburg, participants filled out an informed consent, screening questionnaires, general intelligence screening (APM), and questionnaire on their computer gaming

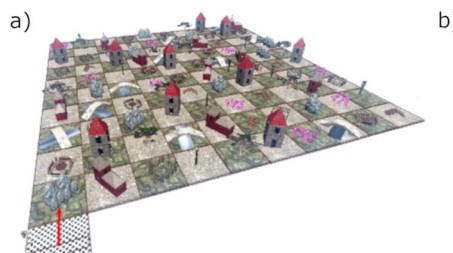 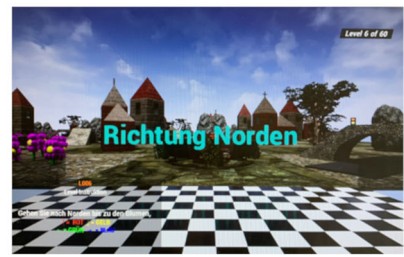

**Fig. 8 3D-Navigation Task. a** Environment of the Navigation Task. **b** Perspective from the starting field. Participants began every level from the starting field outside of the environment. They got an instruction in the left corner of the display to the first goal. The cardinal direction of the current level was shown in the middle of the display only for a few seconds at the beginning. The straight direction was always the given cardinal point. As soon as they reached the first goal the direction to the next goal was given in the left corner.

experience. Furthermore, they were trained in the tasks for the MRI scanning sessions. MRI scanning took place at the Neuroscience Institute of the CDK in Salzburg.

Each scanning session included an 8 min resting state scan, followed by 35 min of task-based functional imaging. During this period, participants completed 20 navigation trials. Each navigation trial was preceded and followed by a 15 second interstimulus interval, which were intermitted by a 30 seconds period during which participants engaged in a verbal fluency task. During the 15 second interstimulus intervals participants saw a blank black screen. These intervals served as a baseline for the MRI analysis. Data of the verbal fluency task were reported elsewhere[87]. After the navigation task, a high-resolution structural scan and a DWI scan were completed, during which participants watched a movie. Overall, one session took ~70 min to complete.

**MRI-Data acquisition**. MRI-Data were acquired on a Siemens Magnetom Trio Tim 3 Tesla scanner, located at the Christian Doppler Klinik (Salzburg, Austria). High-resolution structural images were acquired using a T1-weighted sagittal 3D MPRAGE sequence (TR = 2300 ms, TE = 2.91 ms, TI delay of 900 ms, FOV 256 mm, slice thickness = 1.00 mm, flip angle 9°, voxel size $1.0 \times 1.0 \times 1.0$ mm, 160 sagittal slices). Functional scans were obtained using a T2*-weighted gradient echo planar (EPI) sequence sensitive to BOLD contrast (TR = 2250 ms, TE = 30 ms, FOV 192 mm, matrix size 192 × 192, slice thickness = 3.0 mm, flip angle 70°, voxel size $3.0 \times 3.0 \times 3.0$ mm, 36 transversal slices parallel to the AC-PC line).

**Preprocessing**. Preprocessing of fMRI data was performed as described in Pletzer et al. (2019). Structural images were segmented and normalized using the computational anatomy toolbox (CAT12). For functional images, the first 6 images of each session were discarded. Images were then despiked using the 3d-despiking procedure as implemented in AFNI (afni.nimh.nih.gov). For further preprocessing SPM12 standard procedures and templates were used, including realignment of functional images, slice-timing, normalization of functional images using the normalization parameters obtained by CAT12, smoothing of normalized images using a 6 mm Gaussian kernel. Additionally, after the realignment step, physiological noise was identified using a biophysically-based model[88]. Via the Functional Image Artefact Correction Heuristic (FIACH)[88], images were filtered and 6 regressors of physiological noise were extracted.

**Activation analyses**. Following preprocessing, a 2-stage mixed effects model was applied. By convolving the duration of the event with the canonical hemodynamic response function implemented in SPM12, we modeled one regressor per navigation category (allocentric-Euclidian, allocentric-landmark, egocentric-Euclidian, egocentric-landmark) in the subject-dependent fixed-effects first-level analysis. Instructions and verbal fluency trials, the 6 realignment parameters and the 6 physiological noise parameters obtained from the FIACH procedure were modeled as regressors of no interest. Autocorrelation correction was performed using an AR(1) model[89] and a high pass filter cutoff was set at 128 seconds. One statistical contrast was defined for each of the 4 regressors of interest to compare BOLD-response during each category to baseline.

The subsequent analysis approach was two-fold. First, region of interest (ROI)-based analyses were performed by extracting principle eigenvariates as measures of BOLD-response from a one-sample *t*-test second-level design including all first-level contrast images. ROIs included the hippocampus, caudate and retrosplenial cortex were defined based on Brodman areas in the Wake Forest University (WFU) Pickatlas toolbox[90]. Eigenvalues were compared between sexes and conditions using linear mixed effects models (compare Statistical analysis section).

Second, differences in brain activation due to sex or condition were explored at the whole-brain level. Contrast images (activation maps) were entered into a flexible factorial design modeling the factors sex, perspective, and strategy as well as their interactions. Session was entered as covariate. Since no menstrual-cycle effects were observed at the whole-brain level, menstrual cycle was not controlled in

whole-brain analyses[53]. Contrasts comparing allocentric vs. egocentric perspective, Euclidian vs. landmark-based strategy, men vs. women as well as the perspective*strategy, sex*perspective and sex*strategy interactions were defined as described by Gläscher and Gitelman (2008). To address whether sex differences were mediated via sex hormone influences, additional flexible factorial designs were created, using estradiol, progesterone, and testosterone, respectively, as additional covariate and modeling their interaction with sex. For all second-level designs, we used an extent threshold of k = 40 voxels, an uncorrected primary threshold of $p < 0.001$ and a secondary peak-level FWE-corrected threshold of $p < 0.05$ (indicated as $p_{FWE}$).

The brain networks were visualized with the BrainNet Viewer (http://www.nitrc.org/projects/bnv/)[91].

**Connectivity analyses**. Connectivity analyses were performed using the CONN-toolbox[92]. Seeds for ROI-to-voxel connectivity analyses were the left and right hippocampus, the left and right caudate, and the left and right retrosplenial cortex, which have been demonstrated to mediate spatial learning strategies in animals[93]. Furthermore, we also included left and right V1 in order to assess whether the sex differences obtained during the current study are specific to navigation areas or reflective of more general differences in perception. The preprocessed functional images underwent linear detrending for white matter (WM) and cerebrospinal fluid (CSF) influences, a band-pass filter (0.008–0.09 Hz) and motion-correction. Voxel-wise connectivity maps for each subject and session were entered into flexible factorial designs modeling the factors sex and condition as well as their interaction and session as a covariate. Again, it was addressed whether sex differences were mediated via sex hormone influences, by creating additional flexible factorial designs using estradiol, progesterone, and testosterone, respectively, as additional covariates and modeling their interaction with sex. Like for activation, we used an extent threshold of k = 40 voxels, we used an uncorrected primary threshold of $p < 0.001$ and a secondary peak-level FWE-corrected threshold of $p < 0.05$ (indicated as $p_{FWE}$).

**Statistics and reproducibility**. We included all 72 participants in the analysis. The number of targets reached (TR), as well as ROI-based activation, were analyzed in the context of a linear mixed effects model (lme) using the *lme* function of *nlme* package (Version 1.1-12) of statistics software R 3.3.2[94]. In all models, the participant number (PNr) was modeled as a random factor and session as a fixed factor to control for learning effects. Furthermore, perspective and strategy are included as fixed effects.

In all models, both, the dependent and continuous independent variables were z-standardized using the scale function. Therefore, the coefficients b of fixed effects in the models represent a standardized effect size based on standard deviations, similar to Cohen's *d*.

To assess sex differences, we followed the following rationale: Menstrual-cycle effects for the ROIs and hormone values were already described in Pletzer et al. (2019) and Scheuringer et al. (accepted). For those variables that did show significant menstrual-cycle modulation, it was assessed whether sex differences vary along the menstrual cycle, by including the interaction term sex*cycle in the model (e.g.: BOLD ~ 1|PNr + session + IQ + sex*cycle). If a significant sex*cycle interaction was observed, post-hoc analyses were performed, comparing men and women separately for each cycle phase. If no sex*cycle interaction was observed in any model, subsequent analyses did not include cycle phase as a factor. To assess sex differences models were run according to the following formulas: BOLD ~ 1| PNr + session + IQ + strategy*perspective*sex.

If a significant sex difference was observed, it was addressed, whether this sex difference was moderated via sex hormones. To that end, hormone levels were added to the model (e.g. LI ~ 1|PNr + session + IQ + sex*hormone). *P*-values for the hormonal analyses and for each ROI were FDR-corrected for multiple comparison.

**Reporting summary**. Further information on research design is available in the Nature Research Reporting Summary linked to this article.

## Data availability

Data for ROI-analyses are openly available at https://osf.io/t3v7z/[95] and http://webapps.ccns.sbg.ac.at/OpenData/. MR-images for whole-brain analyses are available from the corresponding author upon reasonable request.

## Code availability

Scripts for behavioral and ROI-analyses are openly available at https://osf.io/t3v7z/[95].

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

## Acknowledgements

This study was funded by the Austrian Science Fund (FWF; P28261, P32276, W1233-G17) as well as ERC Starting Grant 850953. I.N. was supported by the Doctoral College "Imaging the Mind" (FWF; W 1233-B). We thank all participants for their time and willingness to contribute to this study.

## Author contributions

B.P. designed and made the concept of the study. T.H. was responsible for data acquisition. Analysis of the data was performed by B.P., T.H., I.N., and E.H. T.H. and I.N. drafted the manuscript, which was revised and approved by B.P. All authors agree to be accountable for all aspects of the work in ensuring that questions related to the accuracy and integrity of any part of the work are appropriately investigated and resolved.

## Competing interests

The authors declare no competing interests.
