## [Transparent Peer Review File · Communications Biology]

Reviewers' comments:

Reviewer #1 (Remarks to the Author):

Noachtar, Harris and colleagues investigated how sex difference modulated the brain network engaged during spatial navigation using functional MRI. They also assessed the potential impact of navigation strategies in the recruitment of this network. Unlike other studies, the novelty here is that the authors used, in addition to simple parametric analyses of functional data, connectivity analyses. To resume, they found that the strategy used to navigate in a virtual environment (allocentric versus egocentric) engaged different networks, and that these networks are markedly modulated as a function of sex – the network being more right lateralized for men (suggesting a more spatial strategy), and left-lateralized for women (suggesting a more verbal strategy). Overall, the study is well controlled. The analyses provided are in my opinion appropriate, and the interpretations not overstated.

I have several suggestions which I hope will help the authors to refine their manuscript.

(1) Details regarding the navigation task is kept at its minimum. I'm aware that the behavioral task has been described in a previous study, but the details provided in the manuscript are not enough to understand the virtual environment. Please include at least a figure.

(2) The authors assessed participants with the APM task to estimate QI. I'm aware that QI was not statistically different between men and women, but it would have been better to take this variable in consideration in MRI analyses.

(3) With respect to connectivity analyses, the authors used the hippocampus and the caudate as regions of interest, as these structures have been repeatedly involved in spatial learning, especially in animals such as rats. Other structures are also systematically involved in this type of cognitive activities in animals and humans, in particular the retrosplenial cortex. So, this structure would be also an excellent target for the seed-to-voxel connectivity analyses.

(4) To show the specificity of the networks recruited in navigation, it would have been interesting to use an ROI not situated in a structure known to participate in spatial cognition in general for connectivity analyses. In other words, it seems to me important to show that connectivity of another structures of non-interest are not modulated by the task.

Minor:

(1) I find the section introduction very long. I suggest the authors to be more concise.

(2) In the section introduction, the authors state: "[...] a female advantage for verbal and memory tasks, including face and emotion recognition". To include face and emotion recognition in verbal and memory tasks sounds somewhat odd.

(3) In the different histograms, avoid using commas.

(4) I think that the figures can be improved: (i) please include a medial view to allow the reader to better assess the contribution of medial regions; (ii) I also suggest the authors to use scaled colors to better show the voxels the more significantly activated/involved.

Reviewer #2 (Remarks to the Author):

The aim of this work is to assess the effect of variability factors on the neural bases of navigation

using fMRI. More specifically, the authors studied the effects of perspective (allocentric vs. egocentric), strategy (based on the relative position of landmarks vs. metric) and gender (female vs. male). The interaction between these different factors is reported. The authors propose a ROI based analysis (hippocampus and caudate nucleus) and whole brain analysis. In addition, the authors performed a connectivity analysis using gPPI.

A strength of this study is that it separates the strategy and perspective factors that are often confused in previous studies. It also allows a better characterization of a potential gender effect. It also included a reasonable number of participants.

However, some clarifications are required in my opinion.

1) An explicit illustration of the task is missing. As described, it is difficult to understand how the task was carried out and what the subjects were actually seeing during the different steps of the task and whether there was a baseline task.

2) Regarding the hippocampus and caudate involvement according to the strategy/perspective and gender, the authors hypothesized that the easier a strategy or perspective is for a given gender, the greater the range of activation that should be expected in these regions. This hypothesis should be substantiated. Moreover, they use the inverse argument when discussing some of their results. For example, the stronger activation in men in the inferior frontal gyrus (sex x strategy interaction) is interpreted as reflecting a more effortful (and thus less skills) in men compared to women. Finally, the authors should comment on the absence of hippocampus involvement in any of the navigation conditions. Such activations are very commonly reported in the existing navigation literature.

3) The authors reported a perspective x strategy interaction in the lingual gyrus and interpret this interaction as reflecting the importance of lingual gyrus in landmarks-based navigation. However, plots show that this region is actually deactivated compared to the baseline. The same reasoning applies for sex difference in connectivity: Fig 8 display only negative correlations. How do the authors interpret this result?

4) The 3D rendering of the illustrations is not informative regarding the activations on the medial side of the brain.

Respond to reviewers' comments

First, we would like to thank the reviewers for the insightful comments. In the following, we will respond to each comment. We changed the manuscript according to the suggestions of the reviewers. The changes are highlighted within the manuscript.

Reviewer #1:

(1) Details regarding the navigation task is kept at its minimum. I'm aware that the behavioral task has been described in a previous study, but the details provided in the manuscript are not enough to understand the virtual environment. Please include at least a figure.

We thank the reviewer for pointing this out and have expanded the task description and entered a figure displaying two different perspectives of the navigation task.

(2) The authors assessed participants with the APM task to estimate QI. I'm aware that IQ was not statistically different between men and women, but it would have been better to take this variable in consideration in MRI analyses.

We thank the reviewer for this suggestion. Unfortunately, the flexible factorial designs employed, do not allow the assessment of between-subjects differences while controlling for a covariate that doesn't change across the different test sessions, as is the case for IQ. They also do not allow for the assessment of between-subjects differences while controlling for multiple covariates. Since we do consider a control for the learning effect as essential and did not want to abstain from using session as a covariate, we performed the following analyses to check whether any of our results were attributable to IQ:

- (i) **We checked for changes in the within-subjects effects (perspective, strategy) as well as interaction effects (sex*perspective, sex*strategy), when IQ was entered as an additional covariate and did not observe any changes in the results at the whole-brain level.**
- (ii) **We extracted eigenvariates from all clusters with significant between-subject effects (sex differences) and checked for changes in the sex differences, when entering IQ as a covariate in linear mixed effects models. All sex differences remained significant.**

Accordingly, we concluded that none of the results we observed were attributable to IQ and did not further include IQ as additional covariate in the models.

(3) With respect to connectivity analyses, the authors used the hippocampus and the caudate as regions of interest, as these structures have been repeatedly involved in spatial learning, especially in animals such as rats. Other structures are also systematically involved in this type of cognitive activities in animals and humans, in particular the retrosplenial cortex. So, this structure would be also an excellent target for the seed-to-voxel connectivity analyses.

We thank the reviewer for this valuable suggestion and added the retrosplenial cortex as an ROI to the activation and connectivity analyses.

(4) To show the specificity of the networks recruited in navigation, it would have been interesting to use an ROI not situated in a structure known to participate in spatial cognition in general for

connectivity analyses. In other words, it seems to me important to show that connectivity of another structures of non-interest are not modulated by the task.

Again, we thank the reviewer for this valuable suggestion. Unfortunately, already the whole-brain activation results show that there is no structure that is not either activated or deactivated by this task, which is not surprising due to the high task-complexity. We thus did include V1 in the connectivity analyses, as a brain area that is not specifically known for its involvement in spatial cognition, although visual perception obviously plays an important role in spatial processing. Indeed the connectivity analyses show that V1 connectivity is both modulated by the task-factors perspective and strategy, as well as by sex. We agree with the reviewer, that in the absence of a demonstration that our sex differences are specific to brain connections involved in the task, we cannot conclude that the sex differences we observe are task-specific. However, we do believe that these differences are of relevance either way, because even sex differences in resting state connectivity might predispose subjects to process the content of the task in a specific way.

Minor:

(1) I find the section introduction very long. I suggest the authors to be more concise.

We have substantially revised the introduction to make it more concise.

(2) In the section introduction, the authors state: “[...] a female advantage for verbal and memory tasks, including face and emotion recognition”. To include face and emotion recognition in verbal and memory tasks sounds somewhat odd.

We thank the reviewer for pointing this out and have re-formulated the sentence.

(3) In the different histograms, avoid using commas.

All histograms were revised accordingly.

(4) I think that the figures can be improved: (i) please include a medial view to allow the reader to better assess the contribution of medial regions; (ii) I also suggest the authors to use scaled colors to better show the voxels the more significantly activated/involved.

All figures were revised accordingly.

Reviewer #2:

1) An explicit illustration of the task is missing. As described, it is difficult to understand how the task was carried out and what the subjects were actually seeing during the different steps of the task and whether there was a baseline task.

We thank the reviewer for pointing this out and have expanded the task description and entered a figure displaying two different perspectives of the navigation task.

2) Regarding the hippocampus and caudate involvement according to the strategy/perspective and gender, the authors hypothesized that the easier a strategy or perspective is for a given gender, the

greater the range of activation that should be expected in these regions. This hypothesis should be substantiated. Moreover, they use the inverse argument when discussing some of their results. For example, the stronger activation in men in the inferior frontal gyrus (sex x strategy interaction) is interpreted as reflecting a more effortful (and thus less skills) in men compared to women.

Indeed it is always challenging to determine, whether an area is more strongly activated because it is more strongly involved in a process or whether reduced activation of an area indicates more efficient neural processing, i.e. less effort in a task. Actually, those two explanations are not even mutually exclusive as it is equally hard to determine, whether activation is reduced because less effort is required or activation is reduced because less effort is allocated. We do however agree, that switching between interpretations/expectations might be confusing to the reader, particularly since the background supporting our original hypothesis in the introduction is rather scarce, and both directions could be argued. Accordingly, we do refrain from directional hypothesis in the introduction outlining that the theoretical framework could be interpreted to support either direction, and choose a data-driven approach to the interpretations in the discussions. We do think that the effort-based interpretations are indeed warranted in the case of this study, since the phrasing of the directions predetermines the navigation approach participants have to take. We have reworded several sentences to clarify this.

Finally, the authors should comment on the absence of hippocampus involvement in any of the navigation conditions. Such activations are very commonly reported in the existing navigation literature.

We completely agree that this is somewhat unexpected, particularly since the hippocampus was chosen as a region of interest based on the navigation literature. We added a paragraph in the beginning of the discussion to clarify that the task requirements emphasize the updating and manipulation of spatial information over the encoding and retrieval of spatial information and the task should thus be viewed as a task of spatial working memory rather than spatial learning. Hippocampal activation has mostly been observed in situations of spatial learning, while hippocampal deactivation is common in working memory tasks and may in fact reflect working memory load.

3) The authors reported a perspective x strategy interaction in the lingual gyrus and interpret this interaction as reflecting the importance of lingual gyrus in landmarks-based navigation. However, plots show that this region is actually deactivated compared to the baseline. The same reasoning applies for sex difference in connectivity: Fig 8 display only negative correlations. How do the authors interpret this result?

Negative connectivity is often interpreted as indicating inhibitory influences. Accordingly, we do speculate that the caudate is more inhibited during allocentric navigation in women compared to men and during egocentric navigation in men compared to women and that this inhibition may be involved in the suppression of their preferred reference frame, since the perspective was pre-determined by the instructions. We do however outline that this is a speculative interpretation.

4) The 3D rendering of the illustrations is not informative regarding the activations on the medial side of the brain.

All figures were revised accordingly.

Reviewers' comments:

Reviewer #1 (Remarks to the Author):

I thank the authors to have made significant efforts to mitigate my concerns.

The manuscript is greatly improved.

Reviewer #2 (Remarks to the Author):

I thank the authors for their responses and their efforts to clarify many aspects of their study.

Despite their explanations of the role of working memory in their tasks, I remain surprised by the deactivations they report in the regions classically involved in navigation and that they chose for the right reasons as ROIs. Indeed, the tasks proposed in this study most likely induce an implicit encoding of topography. This encoding is partly based on the deactivated regions. The explanations given seem incomplete to me. I did not see a description of the baseline in the text (cross-fixation, black screen, other?). Could it be that the nature of the baseline has led to these deactivations?

Moreover, the authors do not report in the results that the connectivity of V1, chosen as the reference ROI, is in fact affected by perspective, strategy and gender. These data should be included. It raises the question of the specificity of the results regarding differences in connectivity and their interpretation.

Respond to reviewer's comments

Again, we would like thank the reviewer for the insightful comments. In the following, we will respond to the comments. We changed the manuscript according to the suggestions. The new changes are highlighted within the manuscript in red.

Reviewer #1:

I thank the authors to have made significant efforts to mitigate my concerns. The manuscript is greatly improved.

Thank you very much. We appreciated your helpful feedback!

Reviewer #2:

I thank the authors for their responses and their efforts to clarify many aspects of their study.

Thank you again for your insightful comments. They have greatly helped us to improve the manuscript.

Despite their explanations of the role of working memory in their tasks, I remain surprised by the deactivations they report in the regions classically involved in navigation and that they chose for the right reasons as ROIs. Indeed, the tasks proposed in this study most likely induce an implicit encoding of topography. This encoding is partly based on the deactivated regions. The explanations given seem incomplete to me. I did not see a description of the baseline in the text (cross-fixation, black screen, other?). Could it be that the nature of the baseline has led to these deactivations?

The inter-stimulus interval that was used as a baseline was a simple black screen. Accordingly, we think it is unlikely that the nature of the baseline lead to an underestimation of the activation level of these areas. However, we have added a sentence to the discussion that this can not be completely ruled out. In any case we are confident that the activation level did not affect the pattern of connectivity results in our study since these are correlational in nature and depend on the pattern of changes in the BOLD-signal.

Moreover, the authors do not report in the results that the connectivity of V1, chosen as the reference ROI, is in fact affected by perspective, strategy and gender. These data should be included. It raises the question of the specificity of the results regarding differences in connectivity and their interpretation.

Thank you for this suggestion. We added V1 connectivity to the results section and addressed the question of specificity in the very beginning of the discussion section.